# DafnyBench: A Benchmark for Formal Software Verification

**Chloe Loughridge**[*]
Harvard College
cloughridge@college.harvard.edu

**Qinyi Sun**[*†]
Massachusetts Institute of Technology
wendysun@mit.edu

**Seth Ahrenbach**
seth.ahrenbach@omnifederal.com

**Federico Cassano**
Northeastern University
cassano.f@northeastern.edu

**Chuyue Sun**
Stanford University
chuyues@stanford.edu

**Ying Sheng**
Stanford University
ying1123@stanford.edu

**Anish Mudide**
Massachusetts Institute of Technology
amudide@mit.edu

**Md Rakib Hossain Misu**
University of California Irvine
mdrh@uci.edu

**Nada Amin**
Harvard University
namin@seas.harvard.edu

**Max Tegmark**
Massachusetts Institute of Technology
tegmark@mit.edu

## Abstract

We introduce DafnyBench, the largest benchmark of its kind for training and evaluating machine learning systems for formal software verification. We test the ability of LLMs such as GPT-4 and Claude 3 to auto-generate enough annotations for the Dafny formal verification engine to successfully verify over 750 programs with about 53,000 lines of code. The best model and prompting scheme achieved 68% success rate, and we quantify how this rate improves when retrying with error message feedback and how it deteriorates with the amount of required code and annotations. We hope that DafnyBench will enable rapid improvements from this baseline as LLMs and verification techniques grow in quality.

## 1   Introduction

Rapidly improving Large Language Models (LLMs) [1–3] are helping accelerate software development through program synthesis tools. But how can we ensure that LLM-generated code meets our specifications and reliably does precisely what it is supposed to do? Indeed, this remains a persistent problem even with human-written code: major code-testing efforts failed to prevent e.g. bugs causing an Ariane-V rocket explosion [4] and security vulnerabilities in ssh [5] and the Bash shell [6].

Although *formal verification* can guarantee reliability, providing rigorous mathematical proof that software meets specification, it has yet to gain widespread adoption. Formally verifying code is often

---

[*]Equal contribution. Order determined alphabetically.
[†]Corresponding author.

38th Conference on Neural Information Processing Systems (NeurIPS 2024).

a significant burden on the developer [7, 8]. Also, existing formal verification tools involve a major learning curve above and beyond just coding, greatly reducing the pool of people able to do this work.

Machine learning methods have the potential to minimize a common pain point of formal methods, i.e., writing formal specifications. To support automation of formal verification, this paper's goal is to build a benchmark by assembling a suite of formally verified programs written in *Dafny*, a formal verification language developed for easy adoption by programmers due to its similarity with popular imperative programming languages such as Python [9]. For formal verification to succeed, most of these programs require supplementary "annotations" to guide the automated theorem prover.

## 2    Related Work

As summarized in Table 1 below, there is a striking lack of training data for formal verification: while there are hundreds of thousands of training examples for proving mathematical theorems and over ten thousand training examples for synthesizing programs, there are only $66 + 153 = 219$ for proving program correctness. This motivates our work in the current paper to expand the largest existing formal verification benchmarks from *Clover* [10] and *dafny-synthesis* [11].

Table 1: Summary of popular machine learning benchmark datasets for proving mathematical theorems, synthesizing programs, and formally verifying programs. Size is measured by the number of samples in each dataset. In the formal reasoning datasets, each sample is usually a math problem or a theorem. In the program synthesis and verified software programming benchmarks, each sample is a program.

| Category | Dataset | Size |
|---|---|---|
| **Mathematical theorem proving** | CoqGym [12] | 71,000 proofs |
| | LeanDojo [13] | 98,734 proofs |
| | PISA [14] | 138,000 proofs |
| | Natural Proofs [15] | 15,000 proofs |
| | Archive of Formal Proofs [16] | 1 million lines of code |
| **Unverified program synthesis** | APPS [17] | 10,000 programs |
| | HumanEvalX [18, 19] | 165 programs |
| | MBPP [20] | 974 programs |
| | SWEBench [21] | 2,294 programs |
| | LiveCodeBench [22] | grows weekly |
| **Formal software verification** | Clover [10] | 66 programs |
| | Dafny-synthesis [11] | 153 programs |

## 3    DafnyBench Construction

### 3.1    Sourcing Ground Truth Programs

In total, our DafnyBench benchmark contains 782 `ground_truth` stand-alone Dafny programs that compile and verify. These programs come from the following sources:

- **GitHub Scrape**: We scraped all publicly available Dafny files on GitHub published on and before the end of 2023. We adapted a deduplication script from [23] to retain a unique set of the scraped Dafny files. The deduplication process reduced the number of `.dfy` files from ~15,000 to ~5,000. We removed any files that did not verify, which left 1,112 files. We found 374 of these files lacked `ensures` statements (postconditions) and 459 of them lacked `assert` and `invariant` clauses (annotations), and removed the union of these sets, which left 556 `ground_truth` files. Out of these files, 113 verify without any annotations.

- **Clover**: We added 62 ground truth textbook Dafny programs provided by the *Clover* benchmark [10]. Out of these files, 23 verify without any annotations.

- **Dafny-synthesis**: Finally, we included 164 Dafny programs provided by the *dafny-synthesis* benchmark. These problems have been translated from the MBPP benchmark [11]. Out of these files, 72 verify without any annotations.

The `ground_truth` programs in our dataset have on average 2.12 methods, 1.03 functions, and 1.40 lemmas. This places the mean complexity of our examples at a level higher than *Clover* [10] alone, which has only one stand-alone method per example. For more detailed summary statistics of DafnyBench dataset, see Appendix A.

## 3.2 Task Design: Fill Annotations

We implemented the `fill_annotations` task. For this task, we took a `ground_truth` program, removed all of its annotations (all of the `assert` and `invariant` statements in the body of the code), and asked LLM to fill annotations back in so that the resulting program could be verified with Dafny.

**Evaluation Metric** An LLM's attempt to fill annotations back in for a test program is counted as a success if all following conditions are satisfied: 1) The reconstructed program is verified with Dafny; 2) LLM preserves all preconditions (`requires` statements) and postconditions (`ensures` statements); and 3) LLM does not use `{:verify false}` or `{assume false}` to "cheat."

```
method LinearSearch<T>(a: array<T>, P: T -> bool) returns (n: int)
    ensures 0 <= n <= a.Length
    ensures n == a.Length || P(a[n])
    ensures forall i :: 0 <= i < n ==> !P(a[i])
{
    n := 0;
    while n != a.Length
        invariant 0 <= n <= a.Length
        invariant forall i :: 0 <= i < n ==> !P(a[i])
    {
        if P(a[n]) {
            return;
        }
        n := n + 1;
    }
}
```

Figure 1: A verified `ground_truth` program. To create `fill_annotations` task, we remove the `invariant` lines, and ask LLM to fill back in equivalent lines so that the resulting program verifies.

## 4 Experiments

### 4.1 Hyperparameters & Prompts

We set `max_tokens = 4096`, which corresponds to the lowest max output token limit among all the evaluated models, and we set `temperature = 0.3`. We gave each model up to $n = 10$ attempts at a given file. If the model failed on any of the intermediate attempts, it received the Dafny error message and was asked to fill in annotations again with the error message taken into consideration. If it failed on all $n$ attempts, it was considered to fail on that specific test program. See Appendix C for prompts.

### 4.2 Basic Results

We tested GPT-4o, GPT-4 Turbo [24], GPT-3.5 Turbo [25], Claude 3 Opus [2], and CodeLlama-7b-Instruct-hf [26] on the 782 programs. Table 2 shows that Claude 3 Opus achieved the highest success rate $\sim 68\%$.

### 4.3 Difficulty Utilizing Dafny Error Messages

Figure 2 shows how the cumulative success rate improved with more attempts $n$. We see that the best models succeeded on the first try about 54%, with rapidly diminishing returns after that, approaching a plateau at about 65% for $n \sim 5$. This suggests that the LLMs are not great at taking Dafny error messages into consideration, or struggle to cope with the underlying task.

| Model | % Success |
|---|---|
| No LLM | 26.9 |
| GPT-3.5 Turbo | $44.0 \pm 1.8$ |
| GPT-4 Turbo | $59.8 \pm 1.8$ |
| GPT-4o | $59.3 \pm 1.8$ |
| Claude 3 Opus | $\mathbf{67.8} \pm 1.7$ |
| CodeLlama-7b-Instruct-hf | $28.0 \pm 1.6$ |

Table 2: Models' success rates at writing annotations for DafnyBench, with $n = 10$ attempts given. Dafny succeeds in auto-verifying some programs even without annotations, corresponding to the "No LLM" $26.9\%$ success rate baseline.

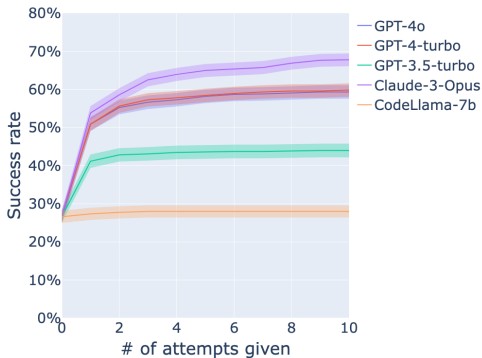

Figure 2: Success rate vs. number of attempts given.

### 4.4 Difficulty Grows with Program Length

Figure 3a shows that the success rate drops with program length. An obvious explanation could be that there is more to verify. Also, as a program gets longer, there may be more dependencies among variables, functions, methods, and classes, increasing the overall verification difficulty level.

### 4.5 Difficulty Grows with Annotation Quantity

Figure 3b shows that the success rate drops with annotation quantity, defined as the number of characters in the lines of annotations. In other words, the success rate drops with the amount of work that LLM needs to do (the amount of text that it needs to insert in the right places).

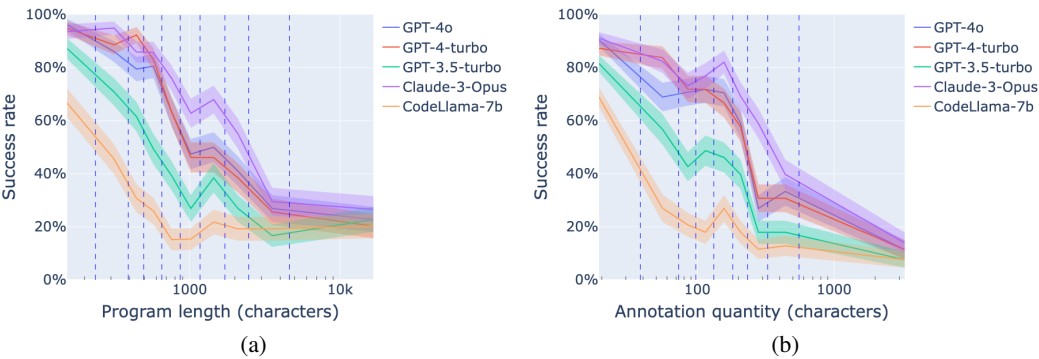

(a)                                    (b)

Figure 3: **Mean success rate of each bin vs. program length (a)**, and **mean success rate of each bin vs. annotation quantity (b)**. The vertical lines indicate the bin boundaries used, where the bins have an almost uniform distribution of the programs. Note the bins are different for the two metrics. For visual clarity, the scales are adjusted for both plots and their $x$-axes do not start at 0 character.

### 4.6 Models' Common Failure Types

To analyze where LLMs failed on the benchmark, we categorized failures into nine types, including verification logic error, code logic error, type error, resolution error, syntax issue, altered specification,

timeout, trivial verification, and others. For a test program that a model failed at, we: 1) checked for timeout, cheating by altering specification, and cheating by trivial verification; and 2) passed Dafny error message from the failed program to Claude and asked it to classify the failure type. Table 3 explains each failure type, and Figure 4 gives by-model statistics of failure types.

Table 3: **Examples of failure types**. Note that the examples are samples, not a complete list, for each failure type.

| Failure Type | Examples |
| --- | --- |
| Code logic error | Index out of range / Target object might be null |
| Verification logic error | Cannot prove termination / Assertion might not hold |
| Syntax issue | lbrace/rbrace expected / Semicolon expected / Unresolved identifier |
| Type error | Value does not satisfy the subset constraints of 'nat' |
| Resolution error | Boogie program had... resolution errors |
| Timeout | Verification timeout |
| Trivial verification | Cheating by using `{:verify false}` or `assume false` |
| Altered specification | Cheating by altering provided specification |
| Other | Failure type not belonging to any listed category above |

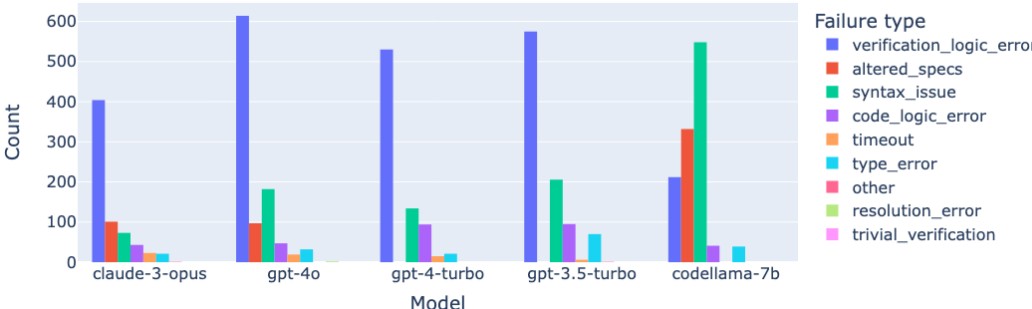

Figure 4: **Counts of failures by failure type and by model**. Note that a model could have multiple failures for a single test program (for example, it might have both verification logic error and syntax issue). Also note that the closed-source models had most of their failures at verification logic, while the open-source model had most of its failures at syntax issues and cheating by altering specification.

## 5 Discussion & Conclusions

We have assembled the largest machine learning benchmark to date for formal software verification and made it publicly available on GitHub at `https://github.com/sun-wendy/DafnyBench`.

### 5.1 Benchmark Evaluation Limitations

Data contamination emerges as a potentially significant limitation for evaluating LLMs on Dafny-Bench. Scraping data from platforms such as GitHub introduces risks of leveraging previous models' training data into the benchmark evaluation, potentially inflating the abilities of certain models.

Another limitation emerges in that DafnyBench does not assess a model's competence in translating natural language into concise formal specifications. Arguably, this conversion is a demanding and crucial skill we seek from language models: the capacity to validate, beyond merely verifying code. The pivotal question is whether a model can assist in identifying the essential properties an algorithm must fulfill. This provides an exciting frontier for future work, which we begin to brainstorm in Appendix D.

For further discussion on LLM's potential for auto-verifying program synthesis and synthesizing specifications from natural language, see Appendix E.

**Acknowledgements:** The authors wish to thank Clark Barrett, Rustan Leino, Daniel Windham, David Brandfonbrener, William Byrd, Josh Engels, and Anastasiya Kravchuk for helpful discussions.

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

## A  Summary Statistics of DafnyBench Dataset

Table 4: Mean and maximum values that describe attributes of a DafnyBench test program.

|  | Mean | Max |
| --- | --- | --- |
| # Methods | 2.12 | 42 |
| # Functions | 1.03 | 42 |
| # Lemmas | 1.40 | 35 |
| # Characters | 1916.47 | 28736 |
| # Annotation characters | 261.23 | 6019 |

# B   Overview of Evaluating LLM on DafnyBench

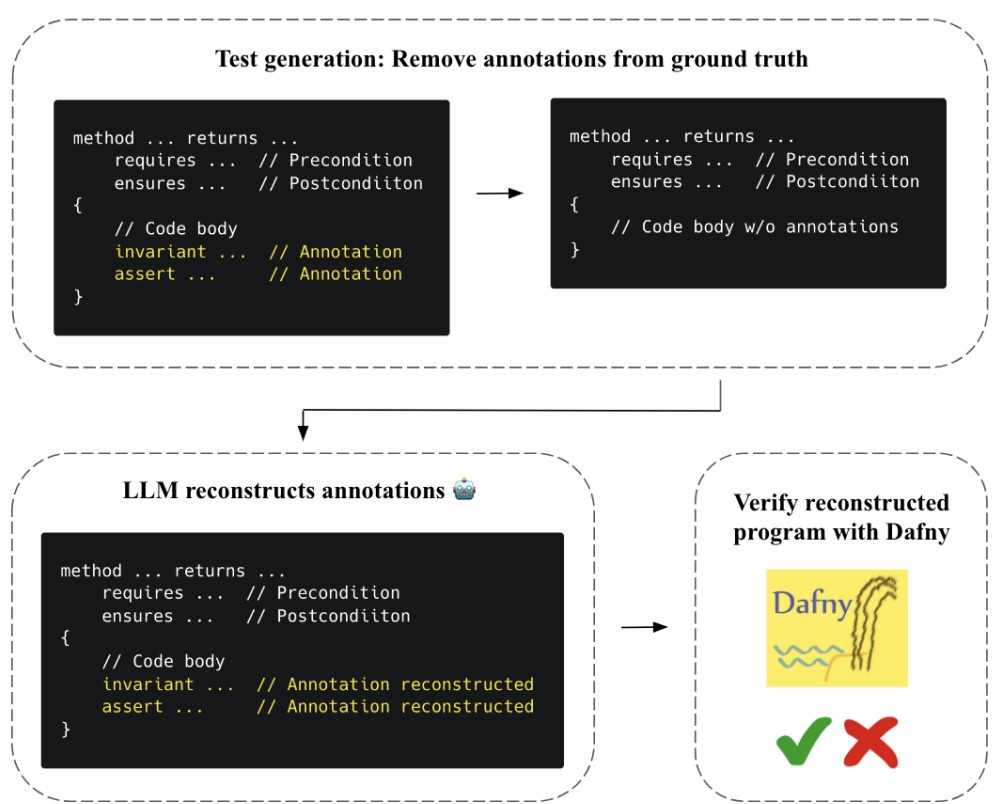

Figure 5: Overview of evaluating LLM on a DafnyBench test program.

# C   Prompt Engineering for Annotation Reconstruction

We based our prompts on the prompts used in the *Clover* benchmark [10], one of the previously largest such benchmarks, since they provide a fairly rigorous precedent. We tried to keep prompts mostly the same across models in order to reduce the difference between model performances that is caused by prompts. However, the prompts are not fully identical. For example, when we ask LLM to simply return the annotations-filled program without any explanation, Claude 3 tends to add explanations that interfere with Dafny compilation. Thus, we had to adjust some prompts slightly to fit each model's peculiarities.

### C.1 GPT Model Famly Prompts

```
SYSTEM_PROMPT = "You are an expert in Dafny. You will be given tasks dealing
                 with Dafny programs including precise annotations."

USER_PROMPT = "Given a Dafny program with function signature, preconditions,
               postconditions, and code, but with annotations missing.
               Please return a complete Dafny program with the strongest
               possible annotations (loop invariants, assert statements,
               etc.) filled back in. Do not explain. Please use exactly the
               same function signature, preconditions, and postconditions.
               Do not ever modify the given lines. Below is the program:"
```

### C.2 Claude 3 Opus Prompts

```
SYSTEM_PROMPT = "You are an expert in Dafny. You will be given tasks dealing
                 with Dafny programs including precise annotations. You should
                 only return code body in all circumstances. No text is allowed."

USER_PROMPT = "Given a Dafny program with function signature, preconditions,
               postconditions, and code, but with annotations missing.
               Please return a complete Dafny program with the strongest
               possible annotation (loop invariants, assert statements,
               etc.) filled back in. Do not explain or output any text. If
               you have to explain, put all explanations in comments form.
               There should only be code body in your output. Please use
               exactly the same function signature, preconditions, and
               postconditions. Do not ever modify the given lines. Below
               is the program:\n```dafny\n"
```

### C.3 CodeLlama-7b-Instruct-hf Prompts

The prompts for CodeLlama-7b-Instruct-hf are the same as those in C.2.

## D    Proposals for Evaluating Strength of Generated Specifications

The evaluation of models' capability to generate formal specifications might be enhanced by integrating the process with the creation of positive and negative test cases for each Dafny implementation. This approach proposes a reward system where models are evaluated based on the number of positive test cases their formal specifications support and the number of negative test cases they successfully reject. However, this method introduces a new challenge: ensuring the test cases accurately reflect the comprehensive meaning intended in the natural language descriptions. The consistency and validity of these test cases become critical, raising questions about the methods used to generate and verify them.

## E    Further Discussion

### E.1    Opportunities for Larger Benchmarks

It will be valuable to further expand formal verification benchmarks, which still remain more than two orders of magnitude smaller than corresponding benchmarks for mathematical theorem proving. One convenient way to expand the number of available problems may involve incorporating Dafny programs from GitHub that have dependencies spread across multiple files (while DafnyBench encompasses increasingly complex multi-step programs, its programs each fit in a single file, avoiding the intricacies associated with distributed files or the integration of external libraries).

Perhaps models that perform especially well on this initial benchmark can later be used to expand it by translating existing Python benchmark problems into Dafny, Rust [27] or other popular formal verification languages.

A subset of the programs we scraped from GitHub do not have appropriate docstrings. By building a benchmark with better code documentation, models may be able to leverage helpful contextual information to better constructing verification annotations.

## E.2 Opportunities for Improved LLM Results

We evaluated the models with a fixed temperature setting and a max output token limit of 4096, and we used prompts that were manually but not very systematically tuned for effectiveness (see Appendix C) — all of these choices probably leave room for improvement.

We do not yet provide an official training dataset or models custom-trained to do well on the DafnyBench evaluation set. However, we do provide the full json file produced by the GitHub scrape, and we separately provide the names of the files we use for the evaluation benchmark. Hence, it is possible for researchers to use files from the Github scrape that are not used in the benchmark as training data, though we cannot at this time provide strong guarantees on similarity between such training problems and the benchmark problems.

We also see opportunities for LLM-related innovation on the algorithmic side: out-of-the-box LLMs provide a floor but not a ceiling for possible performance on this benchmark. For example, fine-tuning or search-based inference-time algorithms might boost models' performances on this benchmark [28].

## E.3 The Potential of Better LLM-Powered Verifiers

LLMs also have potential to improve formal verification in more profound ways than mentioned above, when used in combination with other AI tools. For example, they can help automate the identification of sub-goals and annotations, reducing the search space for automated theorem provers and SAT solvers. A software developer is likely able to specify the high level assurance properties of a piece of code, but may lack familiarity with the complexities of proof sub-goals and annotations. LLMs offer a way to bridge this gap between software developers and formal verification.

Bigger, more general benchmarks can be used to train LLMs to specify sub-goals and annotations in formats most useful to the presently available provers and solvers. Benchmarks covering broad ground, from cryptography, lambda calculus, embedded systems, and avionics, in a variety of widely used programming languages suitable for verification, will help create LLMs that can take real-world software, automatically process and serve it to verification tools, and inform the developer in near real time about the correctness of the code. The problem is analogous to that solved by existing automated theorem provers and model checkers in the domain of mathematics. For a survey on the application of deep learning to automated theorem proving, see [29].

## E.4 The Potential of Auto-Verifying Program Synthesis

Above we discussed the challenge of verifying existing pre-programs. Anther potential of LLMs is use program-synthesis techniques that produce both programs and proofs of their correctness, all at the same time. This makes intuitive sense, since when a human programmer writes code, they typically have an informal proof in their head for why this code is correct. In other words, in addition to bridging the gap from low level implementation to high level specification in the upward direction, LLMs can offer assistance in generating provably correct low level code from high level specifications via program synthesis.

Current approaches to program synthesis enable engineers to encode a desired specification in a high level language, and then through a (hopefully) verified correct compiler generate correct low level code in a language like VHDL [30] or Verilog [31] for hardware synthesis. Program synthesis is limited by the need for a special purpose language or compiler to be constructed and verified correct in its own right. For example, ReWire, a domain specific language defined as a subset of Haskell [32], was manually verified correct using the Coq Interactive Theorem Prover. In order to add a new high-to-low path, a new language or compiler will need to be defined and verified. If an engineer needs to synthesize correct Verilog rather than VHDL, they would likely need to first learn Caisson [33].

LLMs offer a way to generalize this approach. Starting with a high level language, an engineer might be able to specify a system and then leverage a LLM to generate low level code with the

corresponding loop invariants, weakest pre-conditions, strongest post-conditions, etc, included. Early results indicate that an LLM that is able to converse with a human when producing a program can reduce the error rate against a simple programming benchmark by half [20]. If instead of receiving feedback from a human, the LLM were to interact with a suite of formal verification tools, we expect further improvements. The LLM should be capable of generating code that is appropriately annotated for theorem proving, which is exactly the skill assessed by test benches like that described here.

## F    The Minhash Deduplication Algorithm

We can think about deduplicating a set of files by finding groups of "similar"files and then choosing only one file representative from each group to form our final deduplicated set of files. To do this, we can use the Jaccard similarity metric to decide whether one document is a duplicate of another.

The Jaccard similarity metric provides a way to quantify the similarity of two sets. It is defined as [34]:

$$J(A, B) \;=\; \frac{|A \cap B|}{|A \cup B|} \;.$$

In the application to code files, we could consider each file to be a set of $n$-grams, where an $n$-gram is defined as a sequence of $n$ adjacent symbols in a particular order [35], and then apply the Jaccard score as a similarity metric for our files. To directly calculate this Jaccard score, we would need to run string comparison on every $n$-gram, which would have time complexity $O\left(nm^2\right)$ if we have $n$ $n$-grams each with max length $m$ characters. This turns out to be an inefficient method for representing each code file as a set. Instead, the minhash deduplication algorithm approximates the Jaccard similarity between two documents by shingling the documents and comparing the minhash representation of each set of shingles (i.e. we compare fingerprints of documents instead of full documents). The minhash representation of a document is a way to represent a text document as a set of numbers that is faithful to the structure of its content but with a fixed set size that is smaller than the total number of $n$-grams in the document (i.e. the minhash representation of the document is a form of numerical fingerprint of the document). In Figure 6 below, we provide the pseudocode for the minhash algorithm used, based entirely on the script in [23]:

Note that the probability two files have the same min hash value under the same hash function is equivalent to their Jaccard similarity. Concretely, for file $A$ and file $B$:

$$\Pr\left[\,\min h_i(A) = \min h_i(B)\,\right] \;=\; J(A, B)$$

where $\min h_i()$ denotes taking the minimum hash value under hash function $h_i$. This makes sense because, assuming negligible hash collision, $\Pr\left[\min h_i(A) = \min h_i(B)\right]$ is equivalent to the probability that the first $n$-gram hash of $A$ under $h_i$ is equal to the first $n$-gram hash of $B$ under $h_i$. If $h_i$ is a good hash function, then it uniformly distributes the hash values of the original n-gram hashes over the range of $h_i$. Let $c$ denote the number of $n$-grams with equivalent hashes; let $a$ denote the number of $n$-grams from $A$ with smaller hash values than the hash value of corresponding $n$-gram from $B$; let $b$ denote the reverse of the previous category. Then, $\Pr\left[\min h_i(A) = \min h_i(B)\right] \;=\; \frac{c}{a+b+c}$, given the uniformity of $h_1$. Note that $\frac{c}{a+b+c} \;=\; \frac{|A \cap B|}{|A \cup B|} \;=\; J(A, B)$.

## G    Repositories of Scraped Dafny Code

We provide a full list of all repositories whose data we used in the scraped portion of DafnyBench in Tables 5, 6, 7. When reporting the license information, "Renamed so N/A" implies that the original repository we scraped in December 2023 no longer exists under that name. Otherwise, the repositories have either Microsoft open-source licenses, MIT licenses, GNU General Public License v3.0 licenses, Creative Commons Zero v1.0 Universal, Apache 2.0 licenses, or "Other" (which is secretly an MIT License in a strange format, which has been checked manually). In light of this, we release our derivative DafnyBench repository under an Apache 2.0 license and a GNU General Public License v3.0. We note explicitly here that all files from repositories with the Apache 2.0 license have been modified from their original form.

```
function minhash_deduplication(documents, num_permutations, threshold)
    :
    # Preprocess the documents
    for each document in documents:
        tokenize the document into n-grams (shingles)
        hash each n-gram using a hash function (e.g., xxHash or SHA-1)
        store the hashed n-grams in a set

    # Generate permutations
    for i from 1 to num_permutations:
        generate random coefficients a and b
        create a permutation function: (a * x + b) % prime_modulus

    # Create minhash signatures
    signatures = []
    for each document in documents:
        signature = []
        for each permutation function:
            min_hash = INFINITY
            for each hashed n-gram in the document:
                permuted_hash = apply permutation function to hashed n
                    -gram
                min_hash = min(min_hash, permuted_hash)
            append min_hash to signature
        append signature to signatures

    # Perform Locality-Sensitive Hashing (LSH)
    # We use 250 permutations, so to achieve Jaccard similarity
        threshold of 0.5
    # We really only need one band (i.e. one hash table)
    num_bands = choose number of bands
    rows_per_band = num_permutations / num_bands
    candidate_pairs = []
    for each band:
        create an empty hash table
        for each document signature:
            band_signature = subset of signature for the current band
            hash_bucket = hash(band_signature)
            add document to the corresponding hash bucket
        for each hash bucket:
            if number of documents in the bucket > 1:
                generate all pairs of documents in the bucket
                add pairs to candidate_pairs

    # Use a union-find datastructure to track groups of duplicates
    duplicates = UnionFind()
    for each band:
        for each row in hashtable:
            for each hash_bucket:
                if size(hash_bucket) <= 1:
                    continue
                else:
                    cluster_id = min(hash_bucket)
                    for x in hash_bucket:
                        duplicates.union(x, cluster_id)

    # Perform deduplication
    deduplicated_documents = []
    for each document in documents:
        if duplicates.find_root(document) = document:
            add document to deduplicated_documents

    return deduplicated_documents
```

Figure 6: Pseudocode for the minhash deduplication algorithm.

Table 5: Repositories from which DafnyBench utilizes scraped code (no particular order).

| Repository Name | License |
|---|---|
| dafl | No license provided |
| Dafny-Grind75 | No license provided |
| feup-mfes | MIT License |
| Dafny | GNU General Public License v3.0 |
| nitwit | MIT License |
| Dafny-experiences | No license provided |
| Formal_Verification_With_Dafny | No license provided |
| SENG2011 | No license provided |
| M2 | No license provided |
| assertive-programming-assignment-1 | No license provided |
| t1_MF | No license provided |
| dafny-exercise | Other |
| dafny-learn | No license provided |
| software-specification-p1 | No license provided |
| FMSE-2022-2023 | The Unlicense |
| fv2020-tms | No license provided |
| type-definition | No license provided |
| laboratory | No license provided |
| dafny | GNU General Public License v3.0 |
| TFG | GNU General Public License v3.0 |
| SiLemma | MIT License |
| dafny-training | No license provided |
| FormalMethods | No license provided |
| dafny_misc | MIT License |
| vmware-verification-2023 | No license provided |
| CSU55004—Formal-Verification | No license provided |
| MIEIC_mfes | MIT License |
| Dafny-programs | No license provided |
| MFES_2021 | MIT License |
| DafnyPrograms | No license provided |
| cs357 | No license provided |
| formal-methods-in-software-engineering | No license provided |
| Dafny_ProgrammingLanguages | No license provided |
| CSC8204-Dafny | No license provided |
| BPTree-verif | No license provided |
| tangent-finder | No license provided |
| Trab1-Metodos-Formais | No license provided |
| verified-using-dafny | MIT License |
| Metodos_Formais | No license provided |
| lets-prove-blocking-queue | Creative Commons Zero v1.0 Universal |
| Dafny_Programs | No license provided |
| dafny-workout | MIT License |
| Dafny-Projects | No license provided |
| VerifiedMergeSortDafny | No license provided |
| dafny_projects | No license provided |
| pucrs-metodos-formais-t1 | No license provided |
| specTesting | No license provided |
| QS_BoilerPlate1 | No license provided |
| dafny-sandbox | No license provided |
| Formal-Verification | No license provided |
| dafny-duck | No license provided |
| FlexWeek | No license provided |
| 703FinalProject | No license provided |

Table 6: Repositories from which DafnyBench utilizes scraped code (no particular order), continued.

| Repository Name | License |
| --- | --- |
| MFS | No license provided |
| dafny-mini-project | No license provided |
| Software-Verification | No license provided |
| circular-queue-implemetation | No license provided |
| Final-Project-Dafny | No license provided |
| DafnyProjects | No license provided |
| bbfny | No license provided |
| Formal-methods-of-software-development | No license provided |
| Software-building-and-verification-Projects | No license provided |
| software_analysis | No license provided |
| cs245-verification | No license provided |
| dafny-aoc-2019 | No license provided |
| ProjectosCVS | No license provided |
| MFDS | MIT License |
| groupTheory | No license provided |
| dafny-language-server | Other |
| Invoker | Apache License 2.0 |
| formal-verification | No license provided |
| dafny-programs | No license provided |
| ironsync-osdi2023 | Other |
| verified-isort | No license provided |
| paxos_proof | No license provided |
| se2011 | No license provided |
| Dafny_Verify | No license provided |
| Formal-Methods-Project | No license provided |
| 630-dafny | No license provided |
| dafny_examples | MIT License |
| Workshop | No license provided |
| Dafny-Practice | MIT License |
| CVS-handout1 | No license provided |
| CS494-final-project | No license provided |
| iron-sync | Other |
| stunning-palm-tree | Creative Commons Zero v1.0 Universal |
| sat_dfy | No license provided |
| verification-class | MIT License |
| AssertivePrograming | No license provided |
| Dafny-VMC | MIT License |
| libraries | Other |
| cmsc433 | No license provided |
| Correctness | No license provided |
| CVS-Projto1 | No license provided |
| dafleet | MIT License |
| dafny-rope | MIT License |
| protocol-verification-fa2023 | No license provided |
| vfag | No license provided |
| Dafny_Learning_Experience | Apache License 2.0 |
| summer-school-2020 | No license provided |
| BinarySearchTree | Renamed so N/A |
| llm-verified-eval | MIT License |
| Programmverifikation-und-synthese | Renamed so N/A |
| Prog-Fun-Solutions | Renamed so N/A |
| CO3408-Advanced-Software-Modelling-Assignment... | Renamed so N/A |

Table 7: Repositories from which DafnyBench utilizes scraped code (no particular order), continued.

| Repository Name | License |
|---|---|
| DafnyExercises | No license provided |
| test-generation-examples | No license provided |
| HATRA-2022-Paper | No license provided |
| veri-sparse | No license provided |
| Formal-Verification-Project | No license provided |
| formal_verication_dafny | No license provided |
| Simulink-To_dafny | No license provided |
| dafny_experiments | No license provided |
| cs686 | No license provided |
| Program-Verification-Dataset | MIT License |
| Dafny-demo | No license provided |
| dafny-exercises | No license provided |
| metodosFormais | No license provided |
| CS5232_Project | No license provided |
| Dafny-Exercises | No license provided |

# H   Dafny Verification Examples

We take one example test program from DafnyBench, and consider four possible results for the corresponding LLM-reconstructed program: successfully verifies, fails to verify, cheats by including `assume false`, and cheats by including `{:verify false}`. The last three cases are all considered a fail by the DafnyBench evaluation metric.

## H.1   Successful Example

Figure 7 shows a Dafny program that is considered to have successfully verified without cheating.

**Dafny verifier message**: Dafny program verifier finished with 3 verified, 0 errors.

## H.2   Failed Example

Figure 8 shows a Dafny program that fails to be verified.

**Dafny verifier message**: (20,11): Error: index out of range. (30,4): Error: a postcondition could not be proved on this return path. (11,28): Related location: this is the postcondition that could not be proved. Dafny program verifier finished with 2 verified, 2 errors.

## H.3   Cheat Example

Figure 9 shows that a Dafny program cheats by including `assume false`, which DafnyBench evaluation would count as a fail.

**Dafny verifier message**: Dafny program verifier finished with 3 verified, 0 errors.

## H.4   Another Cheat Example

Figure 10 shows that another Dafny program cheats by including `{:verify false}`, which Dafny-Bench evaluation would count as a fail.

**Dafny verifier message**: Dafny program verifier finished with 3 verified, 0 errors.

```
function sorted(a: array<int>) : bool
    reads a
{
    forall i,j : int :: 0 <= i < j < a.Length ==> a[i] <= a[j]
}

method BinarySearch(a: array<int>, x: int) returns (index: int)
    requires sorted(a)
    ensures 0 <= index < a.Length ==> a[index] == x
    ensures index == -1 ==> forall i : int :: 0 <= i < a.Length ==> a[
        i] != x
{
    var low := 0;
    var high := a.Length - 1;
    var mid := 0;

    while (low <= high)
        invariant 0 <= low <= high + 1 <= a.Length
        invariant x !in a[..low] && x !in a[high + 1..]
    {
        mid := (high + low) / 2;
        if a[mid] < x {
            low := mid + 1;
        }
        else if a[mid] > x {
            high := mid - 1;
        }
        else {
            return mid;
        }
    }
    return -1;
}
```

Figure 7: An example response that successfully fills annotations back in and verifies without cheating.

## I  Overdetailed Specification

Figures 11 and 12 show two example programs `update_array_strong.dfy` and `triple_strong.dfy` from the *Clover* benchmark [10], in which the formal specification closely echoes the program implementation.

## J  Ethics Statement

In creating DafnyBench, we took care to use only data that was publicly available on GitHub, and we reference every repository from which we acquired this data, along with their licenses, in Appendix G. Furthermore, we cite the existing verifiable programming benchmarks that we subsume in DafnyBench (i.e. *Clover* [10] and *dafny-synthesis* [11]), and we asked explicit permission from their authors in order to do so. Finally, we cite all models that were used for evaluations on this benchmark [24, 25, 2, 26]. We used these models in accordance with the policies set forth in their API and model card documentation.

## K  Reproducibility Statement

Our benchmark contains the 782 `ground_truth` programs and the corresponding `verification_conditions_removed` programs. Additionally, we include full metadata on all of these files and the evaluation scripts necessary for running the listed models on them. By using the OpenAI and Anthropic APIs, others looking to reproduce this work should not expect to

```
function sorted(a: array<int>) : bool
    reads a
{
    forall i,j : int :: 0 <= i < j < a.Length ==> a[i] <= a[j]
}

method BinarySearch(a: array<int>, x: int) returns (index: int)
    requires sorted(a)
    ensures 0 <= index < a.Length ==> a[index] == x
    ensures index == -1 ==> forall i : int :: 0 <= i < a.Length ==> a[
        i] != x
{
    var low := 0;
    var high := a.Length - 1;
    var mid := 0;

    while (low <= high)
    {
        mid := (high + low) / 2;
        if a[mid] < x {
            low := mid + 1;
        }
        else if a[mid] > x {
            high := mid - 1;
        }
        else {
            return mid;
        }
    }
    return -1;
}
```

Figure 8: An example response that fails to be verified, as it lacks necessary `invariant` statements.

spend more than $300 for a full run of GPT4-o on DafnyBench, $300 for a full run of Claude3 on DafnyBench, $500 for a full run of GPT4-turbo on DafnyBench, and $400 for a full run of GPT-3.5 on DafnyBench. We used the `sglang` package [36] to efficiently query the models. All evaluations were completed on a Linux cluster with an A100 Nvidia GPU.

```
function sorted(a: array<int>) : bool
    reads a
{
   forall i,j : int :: 0 <= i < j < a.Length ==> a[i] <= a[j]
}

method BinarySearch(a: array<int>, x: int) returns (index: int)
    requires sorted(a)
    ensures 0 <= index < a.Length ==> a[index] == x
    ensures index == -1 ==> forall i : int :: 0 <= i < a.Length ==> a[
        i] != x
{
    assume false;
    var low := 0;
    var high := a.Length - 1;
    var mid := 0;

    while (low <= high)
    {
        mid := (high + low) / 2;
        if a[mid] < x {
            low := mid + 1;
        }
        else if a[mid] > x {
            high := mid - 1;
        }
        else {
            return mid;
        }
    }
    return -1;
}
```

Figure 9: An example response that cheats by including `assume false`.

```
function sorted(a: array<int>) : bool
    reads a
{
   forall i,j : int :: 0 <= i < j < a.Length ==> a[i] <= a[j]
}

method {:verify false} BinarySearch(a: array<int>, x: int) returns (
    index: int)
    requires sorted(a)
    ensures 0 <= index < a.Length ==> a[index] == x
    ensures index == -1 ==> forall i : int :: 0 <= i < a.Length ==> a[
        i] != x
{
    var low := 0;
    var high := a.Length - 1;
    var mid := 0;

    while (low <= high)
    {
        mid := (high + low) / 2;
        if a[mid] < x {
            low := mid + 1;
        }
        else if a[mid] > x {
            high := mid - 1;
        }
        else {
            return mid;
        }
    }
    return -1;
}
```

Figure 10: An example response that cheats by including {:verify false}.

```
method UpdateElements(a: array<int>)
  requires a.Length >= 8
  modifies a
  ensures old(a[4]) +3 == a[4]
  ensures a[7]==516
  ensures forall i::0 <= i<a.Length ==> i != 7 && i != 4 ==> a[i] ==
      old(a[i])
{
  a[4] := a[4] + 3;
  a[7] := 516;
}
```

Figure 11: An example program `update_array_strong.dfy` from the *Clover* benchmark [10], in which the formal specification closely echoes the program implementation.

```
method Triple (x:int) returns (r:int)
  ensures r==3*x
{
  r:= x*3;
}
```

Figure 12: Another example program `triple_strong.dfy` from the *Clover* benchmark [10], in which the formal specification closely echoes the program implementation.

