# OpenReview forum: "DafnyBench: A Benchmark for Formal Software Verification"
_NeurIPS.cc/2024/Workshop/MATH-AI — MATH-AI 24_

### Official Review · Reviewer_3g1R · 2024-10-07

**Rating:** 7
**Confidence:** 4

**Review:**

# Summary

The paper introduces DafnyBench, a code generation benchmark consisting of 556 Dafny files from open-source GitHub repositories, and problems from 2 previous Dafny programming benchmarks (62 from CloverBench and 164 from MBPP-DFY). The paper proposes a new task **fill_annotations** that tests models to generate annotations (assert and invariant statements) for Dafny programs so that they can be verified.


# Strengths

## Originality

This is the first benchmark dataset for Dafny code generation derived from real-world code repositories, making it a valuable resource for the community.



## Quality

1. The paper clearly describes the data curation and evaluation processes, including an in-depth analysis of model performance based on program length, annotation quantity, and error distribution.

2. Appendix F provides insightful discussions on potential future work and opportunities, highlighting the paper’s forward-thinking approach.


## Significance

Automated annotation generation is a practically impactful task that can meaningfully aid Dafny programmers.



# Weaknesses


## Significance

As mentioned in Section 5.1 Limitations, while automated annotation generation is already beneficial, generating complete Dafny programs with pre-conditions, post-conditions, and annotations could more comprehensively evaluate the models' capability of Dafny programming.

A potential extension of DafnyBench is to synthesize natural language specifications of the corresponding programs using LLMs. Promising natural language specifications can be filtered using Self-Consistency [1] i.e. give the generated natural language specification to the LLM and check if it generates the correct code.

[[1] Beyond Accuracy: Evaluating Self-Consistency of Code Large Language Models with IdentityChain](https://arxiv.org/abs/2310.14053)


# Questions

1. Could you clarify the rationale behind retaining the 26.9% of programs that Dafny can verify without any annotations?

2. Based on the error distribution analysis in Appendix D, syntax error is the top 3 error source for all models. It's somewhat expected since Dafny is a low-resource language. Could we curate a list of syntax warnings based on those common syntax errors and append it to the prompt so that the model can be evaluated solely on its logical errors? Alternatively, we can provide feedback when we detect syntax errors (or altered specs), inform the model of the specific error, and prompt it to regenerate.

3. Based on the error distribution analysis in Appendix D, altered specs is surprisingly a prominent issue for `gpt-4o` and `claude-3-opus`, but not for `gpt-4-turbo` or `gpt-3.5-turbo`. Typically, the former models are better at instruction following than the latter ones. Do you have any insights as to why this might be the case?

---

### Official Review · Reviewer_yRK6 · 2024-10-07

**Rating:** 7
**Confidence:** 3

**Review:**

**Note to PCs:** The paper is not anonymised. There is a link to a GitHub repo containing the new benchmark, which reveals the authors' names, and the pdf contains list of names in the acknowledgments.

The paper presents a new benchmark for program verification in the Dafny language. The construction follows a standard pipeline: scraping GitHub and appropriately filtering. Different pretrained/fixed LLMs are evaluated on the task of filling in the verification annotations in programs where those annotations have been deleted. The analysis measures the dependence on the error rate on the size of the program and annotations and the distribution of types of errors made by different models.

There is little technical novelty here, but I find it to be a solid benchmark paper and later work can build on it in more extended form. Plenty of directions remain to be explored (fine-tuning, sensitivity to the prompt, syntax-constrained decoding, and so on).

Minor:
- The second sentence of the abstract doesn't make sense to me ('we test the ability to generate enough annotations to successfully verify 750 programs").
- There are some broken `\ref`s.

---

### Official Review · Reviewer_BFUu · 2024-10-07

**Rating:** 8
**Confidence:** 4

**Review:**

This paper introduces DafnyBench, a large benchmark for training and evaluating LLMs for Dafny formal verification.
They provide analyses on how the difficulty of a problem changes and how well LLMs are at solving these problems and taking error feedback from Dafny.
DafnyBench will be helpful for future research on LLM for formal verification.

P.S. There is a non-anonymous acknowledgment section. Not sure if it's problematic @AC.

---

### Official Review · Reviewer_97VT · 2024-10-09
**Review for DafnyBench**

**Rating:** 7
**Confidence:** 3

**Review:**

DafnyBench is a large benchmark designed to evaluate machine learning models' ability to auto-generate annotations for formal software verification. It includes over 750 Dafny programs and tests models GPT-4 and Claude 3, with the latter achieving a 68% success rate. The benchmark measures how well models generate annotations necessary for the Dafny verification engine to confirm program correctness. Success rates decrease with program length and annotation complexity but improve with error feedback. DafnyBench aims to advance formal verification techniques by integrating machine learning.

DafnyBench offers a valuable dataset, providing a valuable resource for testing machine learning models in formal software verification. Its large size and diverse range of programs enable extensive evaluation of annotation generation capabilities. It also encourages the development of improved LLM-based verification techniques through structured benchmarking.

On the downside, the success rates of state-of-the-art models are already high. The benchmark’s reliance on existing data sources introduces potential contamination from models trained on similar datasets. Moreover, it does not assess a model's ability to translate natural language into formal specifications, a key skill in formal verification.

---

### Decision · Program_Chairs · 2024-10-08

Accept